# SPIKEPOINT: AN EFFICIENT POINT-BASED SPIKING NEURAL NETWORK FOR EVENT CAMERAS ACTION RECOGNITION

**Hongwei Ren, Yue Zhou, Xiaopeng Lin, Yulong Huang, Haotian Fu, Jie Song, Bojun Cheng** [*]
The Hong Kong University of Science and Technology (Guangzhou)
{hren066,yzhou883,xlin746 yhuang496,hfu373}@connect.hkust-gz.edu.cn,
{jsongroas,bocheng}@hkust-gz.edu.cn

## ABSTRACT

Event cameras are bio-inspired sensors that respond to local changes in light intensity and feature low latency, high energy efficiency, and high dynamic range. Meanwhile, Spiking Neural Networks (SNNs) have gained significant attention due to their remarkable efficiency and fault tolerance. By synergistically harnessing the energy efficiency inherent in event cameras and the spike-based processing capabilities of SNNs, their integration could enable ultra-low-power application scenarios, such as action recognition tasks. However, existing approaches often entail converting asynchronous events into conventional frames, leading to additional data mapping efforts and a loss of sparsity, contradicting the design concept of SNNs and event cameras. To address this challenge, we propose SpikePoint, a novel end-to-end point-based SNN architecture. SpikePoint excels at processing sparse event cloud data, effectively extracting both global and local features through a singular-stage structure. Leveraging the surrogate training method, SpikePoint achieves high accuracy with few parameters and maintains low power consumption, specifically employing the identity mapping feature extractor on diverse datasets. SpikePoint achieves state-of-the-art (SOTA) performance on five event-based action recognition datasets using only 16 timesteps, surpassing other SNN methods. Moreover, it also achieves SOTA performance across all methods on three datasets, utilizing approximately 0.3% of the parameters and 0.5% of power consumption employed by artificial neural networks (ANNs). These results emphasize the significance of Point Cloud and pave the way for many ultra-low-power event-based data processing applications.

## 1 INTRODUCTION

Event camera is a recent development in computer vision, and it is revolutionizing the way visual information is captured and processed (Gallego et al., 2020). They are particularly well-suited for detecting fast-moving objects, as they can eliminate redundant information and significantly reduce memory usage and data processing requirements. This is achieved through innovative pixel design, resulting in a sparse data output that is more efficient than traditional cameras (Posch et al., 2010; Son et al., 2017). However, most of the event data processing algorithms rely on complex and deep ANNs, which are not aligned with the event camera's low power consumption benefits. Instead, combining event-based vision tasks with SNNs has shown great potential thanks to their highly compatible properties, especially in tasks such as action recognition(Liu et al., 2021a).

Spiking Neural Networks have emerged as a promising alternative that can address the limitations of traditional neural networks for their remarkable biological plausibility, event-driven processing paradigm, and exceptional energy efficiency (Gerstner & Kistler, 2002). The network's asynchronous operations depend on biological neurons, communicating information via precisely timed discrete spikes (Neftci et al., 2019). The event-driven processing paradigm enables sparse but potent computing capabilities, where a neuron activates only when it receives or generates a spike (Hu

---

[*]Corresponding author

et al., 2021). This property gives the network remarkably high energy efficiency and makes it an ideal candidate for processing event-based data. Nevertheless, existing approaches for combining event cameras with SNN require the conversion of asynchronous events into conventional frames for downstream processing, resulting in additional data mapping work and a loss of sparsity (Kang et al., 2020)(Berlin & John, 2020). Moreover, this process leads to the loss of detailed temporal information, which is critical for accurate action recognition (Innocenti et al., 2021). Therefore, developing SNN-compatible novel techniques that operate directly on event data remains challenging.

Point Cloud is a powerful representation of 3D geometry that encodes spatial information in the form of a sparse set of points and eliminates the need for the computational image or voxel conversion, making it an efficient and ideal choice for representing event data (Qi et al., 2017a). The sparse and asynchronous event data could be realized as a compact and informative 3D space-time representation of the scene, bearing a resemblance to the concept of Point Cloud (Sekikawa et al., 2019). Still, Point Cloud networks need frequent high-data dimension transformations and complex feature extraction operators in ANNs. Due to their binarization and dynamic characteristics, these may not function optimally in SNNs.

In this paper, we introduce SpikePoint, the first point-based Spiking Neural Network for effectively and efficiently processing event data in vision-based tasks. Our contributions are as follows: First, we combine event-based vision tasks with SNN by treating the input as Point Clouds rather than stacked event frames to preserve the fine-grained temporal feature and retain the sparsity of raw events. Second, unlike ANN counterparts with a multi-stage hierarchical structure, we design a singular-stage structure harnessing SNN to effectively extract local and global features. This lightweight design achieves effective performance through the back-propagation training method. Lastly, we introduce a pioneering encoding approach to address relative position data containing negative values within the point cloud. This scheme maintains symmetry between positive and negative values, optimizing information representation. We evaluate SpikePoint on diverse event-based action recognition datasets of varying scales, achieving SOTA results on Daily DVS (Liu et al., 2021a), DVS ACTION (Miao et al., 2019), HMDB51-DVS datasets, surpassing even traditional ANNs. Additionally, we attain the SNN's SOTA on the DVS128 Gesture (Amir et al., 2017) and UCF101-DVS dataset (Bi et al., 2020). Notably, our evaluation encompasses an assessment of network power consumption. Compared to both SNNs and ANNs with competitive accuracy, our framework consistently exhibits exceptional energy efficiency, both in dynamic and static power consumption, reaffirming the unequivocal superiority of our network.

## 2 RELATED WORK

### 2.1 EVENT-BASED ACTION RECOGNITION

Action recognition is a critical task with diverse applications in anomaly detection, entertainment, and security. Two primary methods for event-based action recognition are ANN and SNN (Ren et al., 2023). The ANN approaches have yielded several notable contributions, including IBM's pioneering end-to-end gesture recognition system (Amir et al., 2017) and Cannici's asynchronous event-based full convolutional networks (Cannici et al., 2019), while Chadha et al. developed a promising multimodal transfer learning framework for heterogeneous environments (Chadha et al., 2019). Additionally, Bin Yin et al. proposed a graph-based spatiotemporal feature learning framework and introduced several new datasets (Bi et al., 2020), including HMDB51-DVS. (Ren et al., 2023) proposed an effective lightweight framework to deal with event-based action recognition using a tensor compression approach, and (Shen et al., 2023) proposed an efficient data augmentation strategy for event stream data. On the other hand, SNN methods have also demonstrated great potential, with Liu et al. presenting a successful model for object classification using address event representation (Liu et al., 2020) and (George et al., 2020) using multiple convolutional layers and a reservoir to extract spatial and temporal features, respectively. (Liu et al., 2021a) have further advanced the field by extracting motion information from asynchronous discrete events captured by event cameras, and (Yao et al., 2023) developed the Refine-and-Mask SNN (RM-SNN), characterized by its self-adaptive mechanism to modulate spiking responses based on data input. While these SNNs have improved in terms of efficiency, however, their accuracy still falls short when compared to ANN-based approaches.

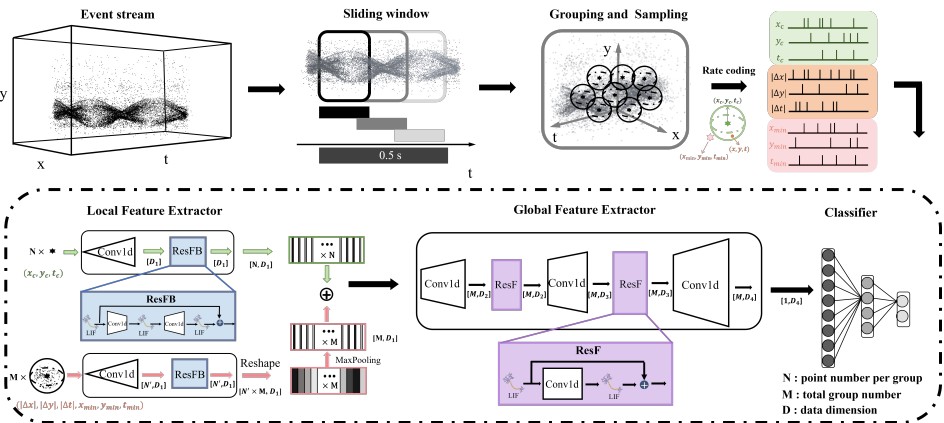

Figure 1: The overall architecture of SpikePoint. The raw event cloud is segmented by the sliding window. Then, the global Point Cloud is transformed into $M$ groups by grouping and sampling. The coordinate is converted into spikes by rate coding, and the results of action recognition are obtained by the local feature extractor, global feature extractor, and classifier in turn.

## 2.2 POINT CLOUD NETWORKS IN ANN

Point-based methods have revolutionized the direct processing of Point Cloud data as input, with PointNet (Qi et al., 2017a) standing out as a remarkable example. PointNet++ (Qi et al., 2017b) took it a step further by introducing a set abstraction module. While it used a simple MLP in the feature extractor, numerous more advanced feature extractors have recently been developed to elevate the quality of Point Cloud processing (Wu et al., 2019; Zhao et al., 2021; Ma et al., 2021; Dosovitskiy et al., 2020; Zhang et al., 2023; Qian et al., 2022; Wu et al., 2023). To apply these methods to the event stream, Wang et al. (Wang et al., 2019) first tackled the temporal information processing challenge while preserving representation in both the x and y axes, achieving gesture recognition using PointNet++. PAT (Yang et al., 2019) further improved this model by incorporating self-attention and Gumbel subset sampling, achieving even better performance in the recognition task. Nonetheless, the current performance of the point-based models still cannot compete with frame-based methods in terms of accuracy. Here, we propose SpikePoint as a solution that fully leverages the characteristics of event clouds while maintaining high accuracy, low parameter numbers, and low power consumption.

## 3 SPIKEPOINT

### 3.1 EVENT CLOUD

Event streams are time-series data that record spatial intensity changes in chronological order. Each event can be represented by $e_m = (x_m, y_m, t_m, p_m)$, where $m$ represents the event number, $x_m$ and $y_m$ denote the spatial coordinates of the event, $t_m$ indicates the timestamp of the event, and $p_m$ denotes the polarity of the event. It is common practice to divide a sample $AR_{\text{raw}}$ into some sliding windows $AR_{\text{clip}}$ by the next formula to facilitate the effective processing of action recognition data.

$$AR_{\text{clip}} = \text{clip}_i \{e_{k \longrightarrow l}\} \mid i \in (1, n_{\text{win}}) \mid t_l - t_k = L \tag{1}$$

where $L$ is the length of the sliding window, $k$ and $l$ represent the start and the end event number of the $i_{\text{th}}$ sliding window, and $n_{\text{win}}$ is the number of the sliding window. To apply the Point Cloud method, four-dimensional events in a clip have to be normalized and converted into a three-dimensional spacetime event $AR_{\text{point}}$. A straight way to do this is to convert $t_m$ into $z_m$ and ignore $p_m$:

$$AR_{\text{point}} = \{e_m = (x_m, y_m, z_m) \mid m = k, k+1, \ldots, l\} \tag{2}$$

With $z_m = \frac{t_m - t_k}{t_l - t_k} \mid m \in (k, l)$, and $x_m, y_m$ are also normalized between $[0, 1]$. After preprocessing, the event cloud is regarded as the pseudo-Point Cloud, which comprises explicit spatial

information $(x, y)$ and implicit temporal information $t$. Through pseudo-Point Cloud, SpikePoint is capable of learning spatio-temporal features which is crucial for action recognition.

## 3.2 SAMPLING AND GROUPING

To unify the number of inputs fed into the network, we utilize random sampling $AR_{\text{point}}$ to construct the trainset and testset. Then, we group these points $PN$ by the Farthest Point Sampling ($FPS$) method and the $K$ Nearest Neighbor ($KNN$) algorithm. $FPS$ is responsible for finding the Centroid of each group, while $KNN$ is utilized to identify $N'$ group members. This process can be abstracted as follows :

$$\text{Centroid} = FPS(PN) \quad \mathcal{G} = KNN(PN, \text{Centroid}, N') \tag{3}$$

The grouping of $PN$ into $\mathcal{G}$ results in a transformation of the data dimension from $[N, 3]$ to $[N', M, 3]$. $N$ is the number of input points, $M$ is the number of groups, and $N'$ is the number of Point Cloud in each group. Given that the coordinates $[x, y, z]$ among the points within each group exhibit similarity and lack distinctiveness, this situation is particularly severe for SNN rate encoding. The standardization method employed to determine the relative position of the point information $[\Delta x, \Delta y, \Delta z]$ with respect to their Centroid is as follows:

$$[\Delta x, \Delta y, \Delta z] = \frac{\mathcal{G} - \text{Centroid}}{SD(\mathcal{G})} \sim N(0, 1), \quad SD(\mathcal{G}) = \sqrt{\frac{\sum_{i=1}^{n}(g_i - \bar{g})^2}{n - 1}} \quad g_i \in \mathcal{G} \tag{4}$$

Where $[\Delta x, \Delta y, \Delta z]$ adheres to the standard Gaussian distribution $N(0, 1)$, $SD$ corresponds to the standard deviation of $\mathcal{G}$ and $g = [x_0, y_0, t_0, \ldots, x_n, y_n, t_n]$. Ultimately, we concatenate the relative position $[\Delta x, \Delta y, \Delta z]$ and Centroid $[x_c, y_c, z_c]$ as the final grouping result.

After grouping, we rate encode the Point Cloud coordinates to meet the SNN network's binary input. It is worth mentioning that the reflected distance information $[\Delta x, \Delta y, \Delta z]$ yields positive and negative values, as shown in Fig. 2. While ANN can successfully handle such information due to their utilization of floating point operations, SNN employs rate coding and binarization, and can't process negative values. Thus, It is necessary to develop a method to effectively handle such information in SNN to achieve accurate results.

A straightforward approach is to normalize $[\Delta x, \Delta y, \Delta z]$ to $[0,1]$, but this can lead to **asymmetric information after passing through points equidistant from the Centroid**, resulting in limited accuracy. The detailed comparison of accuracy is summarized in Table 8 and discussed later. Alternatively, we take the absolute value of the numerator of Eq. 4 to get the $[\Delta|x|, \Delta|y|, \Delta|z|]$ that can perform the spike transformation. However, after such processing, the direction of the relative distance information of the coordinates is lost and the distribution of the response data is changed from the standard normal distribution to the folded normal distribution. The probability density function is:

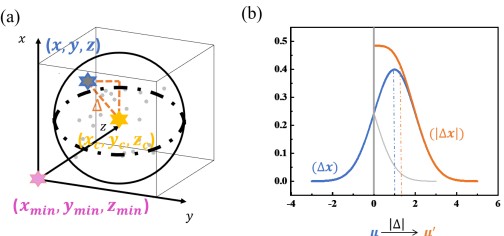

Figure 2: Visualization of our grouping method. (a) The different spatial positions of $[x_c, y_c, z_c]$, $[x_{min}, y_{min}, z_{min}]$ and $[x, y, z]$. (b) The transformation of the distribution after taking absolute.

$$f(x; \mu, \delta^2) = \frac{1}{\sqrt{2\pi}\delta}e^{-\frac{(x-\mu)^2}{2\delta^2}} \rightarrow \frac{1}{\sqrt{2\pi}\delta}(e^{-\frac{(x-\mu)^2}{2\delta^2}} + e^{-\frac{(x+\mu)^2}{2\delta^2}})(x \geq 0) \tag{5}$$

Where $\mu$ and $\delta$ are the mean and standard deviation of the Gaussian distribution, respectively. The expectation $\dot{\mu}$ after the absolute value is given in Eq. 6. $\text{erf}(z) = \frac{2}{\sqrt{\pi}}\int_0^z e^{t^2}dt$ is the error function.

$$\dot{\mu} = \sqrt{\frac{2}{\pi}}\delta e^{-\frac{\mu^2}{2\delta^2}} + \mu \cdot [1 - 2\phi(-\frac{\mu}{\delta})], \quad \phi(x) = \frac{1}{2}[1 + \text{erf}(\frac{x}{\sqrt{2}})] \tag{6}$$

By Eq. 4 - Eq. 6, We maintain codability and shift the expectation of those coordinates to a larger value of $\sqrt{\frac{2}{\pi}}$, which is calculated in Appendix A.1. However, data distribution has also changed

and needs to be compensated. To do so, we replace the input $[\Delta x, \Delta y, \Delta z, x_c, y_c, z_c]$ mentioned above with $[\Delta |x|, \Delta |y|, \Delta |z|], x_{min}, y_{min}, z_{min}], [x_{min}, y_{min}, z_{min}]$ represents the smallest $x, y, z$ values in a specific group. Hence, the increase of the first three input components is compensated by the decrease of the last three input components and the input remains balanced. It is worth noting that the Centroid in each sphere is important and $[x_{min}, y_{min}, z_{min}]$ is not a good indicator of the Centroid, so we introduce a separate branch to extract the global information contained Centroid and do the fusion of these two features in the middle part of the network.

After implementing the aforementioned modifications to the sampling and grouping module, we conducted an analysis that revealed a significant decrease in both the mean relative error (MRE) of rate coding and the coefficient of variation (CV) of the data. This reduction in MRE and CV provides a fundamental explanation for the efficacy of our proposed methodology as evidenced by the results. For the step-by-step derivation of the formulas and the validation of the dataset, please consult Appendix A.2 and A.3.

## 3.3 Singular Stage Structure

The SpikePoint model is unique in its utilization of a singular-stage structure as shown in Fig. 1, in contrast to the hierarchical structure employed by all other ANN-based methods for Point Cloud networks. While this hierarchical paradigm has become the standard design approach for ANNs, it is not readily applicable to SNNs because spike-based features tend to become sparse and indistinguishable as the depth of the stage increases and the training method based on backpropagation has a serious gradient problem. In light of this, we develop **a novel, streamlined network architecture that effectively incorporates the properties of SNNs**, resulting in a simple yet highly efficient model adept at abstracting local and global geometry features. The specific dimensional changes and SpikePoint's algorithms can be referred to in Appendix A.6 and Algorithm A.5.4.

### 3.3.1 Basic Unit

Next, the basic unit of the feature extractor will be introduced in the form of a formula. The first is a discrete representation of LIF neurons. We abstract the spike into a mathematical equation as follows:

$$S_j(t) = \sum_{s \in C_j} \gamma(t - s), \quad \gamma(x) = \theta(U(t, x) - V_{th}) \tag{7}$$

where $S$ represents the input or output spike, $C$ represents the set of moments when the spike is emitted, $j$ is the $j^{th}$ input of the current neuron, $\gamma$ represents the spike function, $\theta$ represents the Heaviside step function, $V_{th}$ denotes the threshold of neuron's membrane potential and $\gamma = 1$ if $U(t, x) - V_{th} \geq 0$ else $\gamma = 0$. We proceed to define the process of synaptic summation in the SNN using the following equation:

$$I[n] = e^{-\frac{\Delta t}{\tau_{\text{syn}}}} I[n - 1] + \sum_j W_j S_j[n] \tag{8}$$

The aforementioned equation delineates the mechanism of neuronal synaptic summation, where the current neuron's input is denoted as $I$, $\tau_{\text{syn}}$ conforms to the synapse time constant, $\Delta t$ represents the simulation timestep, $n$ represents the discrete timestep, while $j$ refers to the antecedent neuron number.

$$U[n + 1] = e^{-\frac{\Delta t}{\tau_{\text{mem}}}} U[n] + I[n] - S[n] \tag{9}$$

where the variable $U$ denotes the membrane potential of the neuron, $n$ represents discrete timesteps, while $\tau_{\text{mem}}$ is the membrane time constant. Moreover, $S$ signifies the membrane potential resetting subsequent to the current neuron transmitting the spike.

SNN is prone to gradient explosion and vanishing during training via backpropagation through time (BPTT), owing to the unique properties of their neurons as described above. Drawing on the utilization of conventional regularization techniques, such as dropout and batch normalization, we endeavor to incorporate a residual module to address the issues of overfitting and detrimental training outcomes. In this work, we achieve identity mapping by modifying the residual module after neurons in Eq. 10 refer (Hu et al., 2021; Fang et al., 2021a; Feng et al., 2022). And the coefficient $\sigma'(I_i^{l+m-1} + S_j^l)$ in Eq. 29 that corresponds to the residual term is canceled out during error

Table 1: Specific information on the five datasets.

| DataSet | Classes | Sensor | Avg.length | Train | Test | Sliding Window | Overlap |
|---|---|---|---|---|---|---|---|
| DVS128 Gesture (Amir et al., 2017) | 10/11 | DAVIS128 | 6.52 s | 26796 | 6959 | 0.5 s | 0.25 s |
| Daliy DVS (Liu et al., 2021a) | 12 | DAVIS128 | 3 s | 2924 | 731 | 1.5 s | 0.5 s |
| DVS Action (Miao et al., 2019) | 10 | DAVIS346 | 5 s | 932 | 233 | 0.5 s | 0.25 s |
| HMDB51-DVS (Bi et al., 2020) | 51 | DAVIS240 | 8 s | 24463 | 6116 | 0.5 s | 0.5 s |
| UCF101-DVS (Bi et al., 2020) | 101 | DAVIS240 | 6.6s | 119214 | 29803 | 1s | 0.5s |

propagation. This coefficient consistently remains below 1, attributed to surrogate gradients. The accumulative multiplication will cause the gradient to disappear during backpropagation. A detailed derivation can be found in Appendix A.4, which describes how this connection solves the problem.

$$S^l = \text{LIF}(I + S^{l-1}) \longrightarrow \text{LIF}(I) + S^{l-1} \tag{10}$$

We have defined two basic units, $ResFB$ and $ResF$, with and without a bottleneck, respectively, as depicted in Fig. 1, and have incorporated them into the overall architecture. Due to the specificity of Point Cloud networks, feature dimensioning usually uses one-dimensional convolution. To maintain the advantage of the residual module, we do not do anything to the features of the residual connection. As a result, the module's input and output dimensions remain the same.

### 3.3.2 LOCAL FEATURE EXTRATOR

The local feature extractor is crucial in abstracting features within the Point Cloud group, similar to convolutional neural networks with fewer receptive fields. As each point is processed, it is imperative to adhere to the principle of minimizing the depth and width of the extractor to ensure the efficiency of the SpikePoint. To this end, we have employed the $ResFB$ unit with a bottleneck to streamline the network design as the following equations, result in more advanced extracted features.

$$X_1 = [\Delta|x|, \Delta|y|, \Delta|z|, x_{min}, y_{min}, z_{min}], X_2 = [x_c, y_c, z_c] \tag{11}$$

$$F_{l1} = ResFB(\text{Conv1D}(X_1)) \tag{12}$$

$$F_{l2} = ResFB(\text{Conv1D}(X_2)) \tag{13}$$

$$F_{\text{local}} = \text{MaxPool}(F_{l1}) + F_{l2} \tag{14}$$

We evaluate both the concat and add operations for the two-channel feature fusion in the ablation experiments section.

### 3.3.3 GLOBAL FEATURE EXTRATOR

The local feature extractor aggregates point-to-point features within each group into intra-group feature vectors, while the global feature extractor further abstracts the relationships between groups into inter-group feature tensors. The input's dimension for the final classifier is intimately linked with the width of the global feature extractor. Thus, it is crucial to enable the feature extractor to expand its width as much as possible while utilizing a limited depth, and simultaneously ensuring the extracted features are of high-level abstraction. The feature extractor can be formulated as follows:

$$L(x) = ResF(\text{Conv1D}(x)) \tag{15}$$

$$F_m = L_2(L_1(F_{\text{local}})) \tag{16}$$

$$F_{\text{global}} = \text{MaxPool}(\text{Conv1D}(F_m)) \tag{17}$$

The final extracted features, denoted as $F_{\text{global}}$, will be transmitted to the classifier A.7.3 for action recognition. To accommodate small and large datasets, we utilized two distinct ascending dimensionality scales while ensuring consistency in our architecture where specific feature dimensions are illustrated through Appendix Fig. 8.

## 4 EXPERIMENT

### 4.1 DATASET

We evaluate SpikePoint on five event-based action recognition datasets of different scales, and more details about the dataset are presented in Appendix A.5. These datasets have practical applications and are valuable for research in neuromorphic computing.

### 4.2 PREPROCESSING

The time set for sliding windows is 0.5 s, 1 s, and 1.5 s, as shown in Table 1. The coincidence area of adjacent windows is set as 0.25 s in DVS128 Gesture and DVS Action datasets, and 0.5 s in other datasets. The testset is 20% randomly selected from the total samples.

## 4.3 NETWORK STRUCTURE

This paper uses a relatively small version network for the Daily DVS and DVS Action, and a larger version network for the other three datasets. It should be emphasized that the architectures are identical, with only a shift in dimensionality in the feature extraction as presented in Appendix A.7.

## 4.4 RESULTS

**DVS128 Gesture:** The DVS128 Gesture dataset is extensively evaluated by numerous algorithms, serving as a benchmark for assessing their efficiency and effectiveness. Compared to the other SNN methods, SpikePoint achieves SOTA results with an accuracy of 98.74% and 98.1% on class 10 and 11, respectively. Comparing the ANN methods, our SpikePoint model demonstrates superior accuracy compared to lightweight ANN such as EV-VGCNN and VMV-GCN, despite the latter employing a

Table 2: SpikePoint's performance on DVS128 Gesture.

| Name | Method | Param | Acc |
|---|---|---|---|
| TBR+I3D (Innocenti et al., 2021) | ANN | 12.25 M | 99.6% |
| PointNet++ (Qi et al., 2017b) | ANN | 1.48 M | 95.3% |
| EV-VGCNN (Deng et al., 2021) | ANN | 0.82 M | 95.7% |
| VMV-GCN (Xie et al., 2022) | ANN | 0.86 M | 97.5% |
| SEW-ResNet (Fang et al., 2021a) | SNN | - | 97.9% |
| Deep SNN(16 layers) (Amir et al., 2017) | SNN | - | 91.8% |
| Deep SNN(8 layers) (Shrestha & Orchard, 2018) | SNN | - | 93.6% |
| Conv-RNN SNN(5 layers)(Xing et al., 2020) | SNN | - | 92.0% |
| Conv+Reservoir SNN (George et al., 2020) | SNN | - | 65.0% |
| HMAX-based SNN (Liu et al., 2020) | SNN | - | 70.1% |
| Motion-based SNN (Liu et al., 2021a) | SNN | - | 92.7% |
| **SpikePoint** | **SNN** | **0.58 M** | **98.74%** |

larger number of parameters, as shown in Table 2. This indicates that Point Clouds' rich and informative nature may confer an advantage over voxels in conveying complex and meaningful information. The current SOTA employed by ANNs utilizes nearly 21 times more parameters than SpikePoint, resulting in a 0.86% increase in performance.

**Daily DVS:** SpikePoint achieves SOTA on this dataset with 97.92% accuracy on the testset, and outperforms all ANN and SNN networks. Notably, it uses only 0.3% of its parameters compared to the high-performing ANN network, as shown in Table 3. Upon visualization of the dataset, we find that it has relatively low noise, and the number of events is evenly distributed in the time domain.

Table 3: SpikePoint's performance on Daily DVS.

| Name | Method | Param | Acc |
|---|---|---|---|
| I3D(Carreira & Zisserman, 2017) | ANN | 49.19 M | 96.2% |
| TANet(Liu et al., 2021b) | ANN | 24.8 M | 96.5% |
| VMV-GCN (Xie et al., 2022) | ANN | 0.84 M | 94.1% |
| TimeSformer (Bertasius et al., 2021) | ANN | 121.27 M | 90.6% |
| HMAX-based SNN (Xiao et al., 2019) | SNN | - | 68.3% |
| HMAX-based SNN (Liu et al., 2020) | SNN | - | 76.9% |
| Motion-based SNN (Liu et al., 2021a) | SNN | - | 90.3% |
| **SpikePoint** | **SNN** | **0.16 M** | **97.92%** |

The action description is well captured by the 1024 points sampled using random sampling. With this in mind, we conduct the ablation study on this dataset, including the processing of negative values (absolute or [0,1]), the utilization of dual channel inputs, and the techniques employed for feature fusion, which are discussed in detail in the Ablation study.

**DVS Action:** SpikePoint also obtains SOTA results on this dataset, as shown in Table 4. In the same way, we visualize the aedat files in the dataset but find that most of the data have heavy background noise, causing us to collect a lot of noise during random sampling. So we preprocess the dataset and finally achieve 90.6% accuracy. This reflects a problem random sampling will not pick up useful points when there is too

Table 4: Model's performance on DVS ACTION.

| Name | Method | Param | Acc |
|---|---|---|---|
| ST-EVNet (Wang et al., 2020) | ANN | 1.6 M | 88.7% |
| PointNet (Qi et al., 2017a) | ANN | 3.46 M | 75.1% |
| Deep SNN(6 layers) (Gu et al., 2019) | SNN | - | 71.2% |
| HMAX-based SNN (Xiao et al., 2019) | SNN | - | 55.0% |
| Motion-based SNN (Liu et al., 2021a) | SNN | - | 78.1% |
| **SpikePoint** | **SNN** | **0.16 M** | **90.6%** |

much background noise. More reasonable sampling will improve the model's robustness.

**HMDB51-DVS:** Compared with the first three datasets which have relatively few categories, HMDB51-DVS has 51 different categories. Our experiments demonstrate that SpikePoint excels not only on small-scale datasets but also demonstrates impressive adaptability to larger ones. As shown in Table 5, SpikePoint outperforms all ANN methods, despite using very few parameters.

Table 7: Evaluating IBM gesture's inference dynamic and static energy consumption.

| Model | Input | Timestep | Accuracy(%) | OPs(G) | Dynamic(mJ) | Para.(M) | Static(mJ) |
|---|---|---|---|---|---|---|---|
| **SpikePoint** | **Point** | **16** | **98.7** | **0.9** | **0.82** | **0.58** | **0.756** |
| tdBN(Zheng et al., 2021) | Frame | 40 | 96.9 | 4.79 | 4.36 | 11.7 | 15.305 |
| Spikingformer(Zhou et al., 2023) | Frame | 16 | **98.3** | 3.72 | 4.26 | **2.6** | **3.401** |
| Spikformer(Zhou et al., 2022) | Frame | 16 | 97.9 | 6.33 | 10.75 | 2.6 | 3.401 |
| Deep SNN(16)(Amir et al., 2017) | Frame | 16 | 91.8 | 2.74 | 2.49 | 1.7 | 2.223 |
| Deep SNN(8)(Shrestha & Orchard, 2018) | Frame | 16 | 93.6 | 2.13 | 1.94 | 1.3 | 1.7 |
| PLIF (Fang et al., 2021b) | Frame | 20 | 97.6 | 2.98 | 2.71 | 17.4 | 22.759 |
| TBR+I3D (Innocenti et al., 2021) | Frame | ANN | **99.6** | 38.82 | 178.6 | **12.25** | **160.23** |
| Event Frames+I3D (Bi et al., 2020) | Frame | ANN | 96.5 | 30.11 | 138.5 | 12.37 | 16.18 |
| RG-CNN (Miao et al., 2019) | voxel | ANN | 96.1 | 0.79 | 3.63 | 19.46 | 25.45 |
| ACE-BET (Liu et al., 2022) | voxel | ANN | 98.8 | 2.27 | 10.44 | 11.2 | 14.65 |
| VMV-GCN (Xie et al., 2022) | voxel | ANN | 97.5 | 0.33 | 1.52 | 0.84 | 1.098 |
| PoinNet++ (Qi et al., 2017b) | point | ANN | 95.3 | 0.872 | 4.01 | 1.48 | 1.936 |

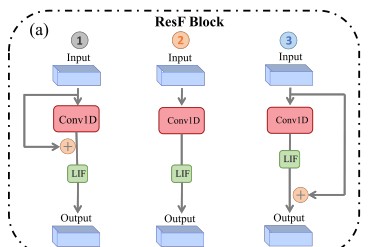 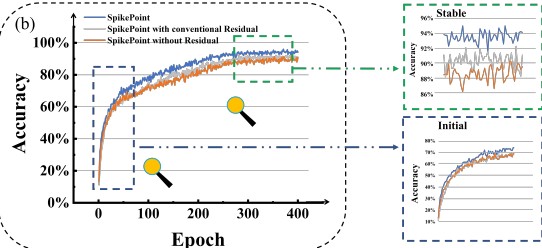

Figure 3: $ResF$ ablation experiment (a) and the result (b) on DVS ACTION dataset.

This indicates that SpikePoint has excellent generalization capabilities. It should be added that the SpikePoint used for DVS Gesture and HMDB51-DVS is both the larger model and the number of parameter differences are only due to the different categories classified by the final classifier. As the final output layer is a voting layer, the difference in parameter quantities is 0.21 M.

**UCF101-DVS:** We continue to test on large-scale datasets with 101 categories, which are challenging to model. As illustrated in Table 6, SpikePoint achieves state-of-the-art results for Spiking Neural Networks with remarkably few parameters, approaching the SOTA results achieved by ANN. These experiments comprehensively demonstrate that SpikePoint is an efficient and effective model.

Table 5: Model's performance on HMDB51-DVS.

| Name | Method | Param | Acc |
|---|---|---|---|
| C3D (Tran et al., 2015) | ANN | 78.41 M | 41.7% |
| I3D (Carreira & Zisserman, 2017) | ANN | 12.37 M | 46.6% |
| ResNet-34 (He et al., 2016) | ANN | 63.7 M | 43.8% |
| ResNext-50 (Hara et al., 2018) | ANN | 26.5 M | 39.4% |
| P3D-63 (Qiu et al., 2017) | ANN | 25.74 M | 40.4% |
| RG-CNN (Bi et al., 2020) | ANN | 3.86 M | 51.5% |
| **SpikePoint** | **SNN** | **0.79 M** | **55.6%** |

Table 6: Model's performance on UCF101-DVS.

| Name | Method | Param | Acc |
|---|---|---|---|
| RG-CNN +Incep.3D (Bi et al., 2020) | ANN | 6.95M | 63.2% |
| I3D (Carreira & Zisserman, 2017) | ANN | 12.4M | 63.5% |
| ResNext-50 (Hara et al., 2018) | ANN | 26.05M | 60.2% |
| ECSNet-SES (Chen et al., 2022) | ANN | - | 70.2% |
| Res-SNN-18 (Fang et al., 2021a) | SNN | - | 57.8% |
| RM-RES-SNN-18 (Yao et al., 2023) | SNN | - | 58.5% |
| SpikePoint | SNN | 1.05M | 68.46% |

## 4.5 POWER CONSUMPTION

Spikepoint's power consumption exhibits substantial advantages over ANNs and SNNs as shown in Table 7. For details on calculating power consumption, please refer to A.9.

## 4.6 ABLATION STUDY

$ResF$ **ablation:** To verify the efficacy of the proposed feature extractor $ResF$, we conduct ablation experiments on the DVS ACTION dataset by varying only the feature extractor variable, keeping the overall architecture, hyperparameters, and datasets consistent. As depicted in Fig. 3(a), three groups of experiments are conducted: the first group utilizes the residual structure with the same architecture as ANN, the second group uses the SpikePoint model without residual connection, and the third group employs the SpikePoint. The results of experiments are marked with blue, orange, and gray, respectively, in Fig. 3(b). The curves demonstrate the superiority of the $ResF$ block, and the results zoom in on the accuracy curves at the beginning and stabilization periods of the model. SpikePoint converges fastest at the beginning of training and has the highest accuracy after stabilization. Compared with the model without residual, it can be proved that this module

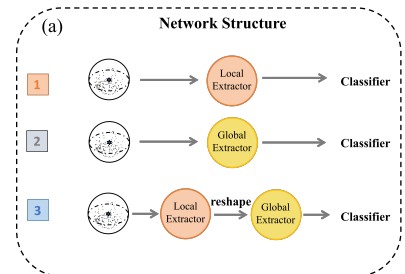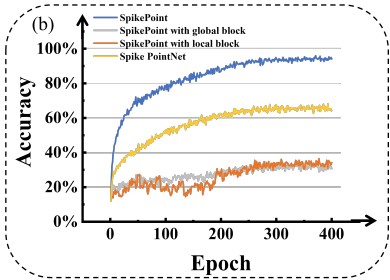

Figure 4: Structural ablation experiment (a) and the result (b) on DVS ACTION dataset.

Table 9: Ablation study of SNN's timesteps.

| Time steps | 2 | 4 | 8 | 12 | **16** | 24 | 32 |
|---|---|---|---|---|---|---|---|
| DailyDVS Acc.(%) | 92.75 | 95.17 | 96.53 | 97.22 | **97.92** | 96.7 | 96.01 |
| DVS Action Acc.(%) | 60.93 | 71.88 | 80.09 | 85.65 | **90.6** | 88.39 | 81.03 |

can improve the model's fitting ability. SpikePoint is better at convergence speed and convergence accuracy than copying the ANN scheme.

**Structural ablation:**

In the structure ablation experiment, we conduct four comparison experiments. The first group involves the standard SpikePoint. The second group is the network with only a local feature extractor. The third group focuses solely on global feature extraction of the Point Cloud set without any grouping or sampling operations, as shown in Fig. 4(a).

Table 8: Ablation study on grouping in Daily DVS dataset.

| No. | Absolute | $[x_{min}...]$ | $[x_c...]$ | Branch | Fusion | Performance |
|---|---|---|---|---|---|---|
| 1 | × | × | ✓ | single | | 97.22% |
| 2 | $[0,1]$ | × | ✓ | single | | 96.53% |
| 3 | $[0,1]$ | ✓ | × | single | | 97.36% |
| 4 | ✓ | ✓ | × | single | | 97.63% |
| 5 | ✓ | × | ✓ | single | | 97.78% |
| 6 | ✓ | ✓ | × | **double** | **Add** | **97.92%** |
| 7 | ✓ | ✓ | × | double | Concat | 97.50% |
| 8 | ✓ | × | ✓ | double | Add | 97.50% |
| 9 | × | × | ✓ | double | Add | 97.22% |
| 10 | $[0,1]$ | × | ✓ | double | Add | 96.25% |

The fourth group is the SNN version of PointNet, which with an architecture that is essentially identical to the third group except for differing dimensions of $[32, 64, 128, 256]$ and $[64, 128, 256, 512, 1024]$, respectively. The results provide evidence of the effectiveness of combining local and global features. As depicted in Fig. 4(b), the gray and orange lines represent architectures with only global and local feature extractors, respectively. Not surprisingly, these models perform poorly on the DVS Action dataset, with an accuracy of only 40%. To compare with PointNet, we adjust the dimensionality of each stage to be consistent with it, and the yellow color indicates the resulting training curve. We observe that the model shows relatively better results when the dimensionality reaches 1024.

**Timestep ablation:** The timestep serves as a crucial hyperparameter, profoundly influencing the accuracy and power consumption in SNN. The choice of the key parameter is a result of the controlled experiment with varying timesteps. We do seven sets of comparison experiments on Daily DVS and DVS Action respectively, and the results are shown as the accuracy of the testset in Table 9.

**Grouping ablation:** To overcome the challenge of rate encoding input with negative values, we introduce five relevant variables, each corresponding to the number of columns in Table 8. We then conducted several experiments to verify their validity. The results demonstrate the effectiveness of taking absolute values for input, which are superior to normalization, with 2 and 5 in Table 8. And $[x_{min}, y_{min}, z_{min}]$ can have a certain corrective effect with 2 and 3, 6 and 8. However, Centroid affects the results more than $[x_{min}, y_{min}, z_{min}]$ in the single channel with 4 and 5. Another branch is used to compensate for this phenomenon with 4 and 6. The dual path structure is better than the single path with 5 and 6, and experiments have shown that the $Add$ operation is better with 6 and 7.

## 5 CONCLUSION

In this paper, we introduce a full spike event-based network that effectively matches the event cameras' data characteristics, achieving low power consumption and high accuracy. SpikePoint as a singular-stage structure is capable of extracting both global and local features, and it has demonstrated excellent results on five event-based action recognition datasets using back-propagation but not converting ANNs to SNNs. Going forward, our aim is to extend the applicability of SpikePoint to other fields of event-based research, such as SLAM and multimodality.

## 6 ACKNOWLEDGEMENT

This work was supported in part by The Hong Kong University of Science and Technology (Guangzhou) Joint Funding Program under Grant 2023A03J0154 and Grant 2023A03J0013, as well as from the Young Scientists Fund of the National Natural Science Foundation of China (Grant 62305278).

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

# A APPENDIX

## A.1 THE SHIFT OF $\mu$

By taking the absolute values, the normalized data becomes an all-positive distribution, and the corresponding PDF is adapted as follows:

$$f(x; \mu, \delta^2) = \frac{1}{\sqrt{2\pi}\delta}(e^{-\frac{(x-\mu)^2}{2\delta^2}} + e^{-\frac{(x+\mu)^2}{2\delta^2}})(x \geq 0) \tag{18}$$

The expectation of the new distribution can be obtained by integrating:

$$\dot{\mu} = \int_0^\infty x \cdot f(x; \mu, \delta^2) dx \tag{19}$$

The data after standardization as obeying the standard Gaussian distribution$\sim$N(0,1), bringing in $\mu = 0$ and $\delta = 1$:

$$\dot{\mu} = \int_0^\infty x \frac{1}{\sqrt{2\pi}}(e^{-\frac{x^2}{2}} + e^{-\frac{x^2}{2}})dx = \sqrt{\frac{2}{\pi}} \int_0^\infty x e^{-\frac{x^2}{2}} dx = \sqrt{\frac{2}{\pi}} \int_0^\infty e^{-\frac{x^2}{2}} d(\frac{x^2}{2}) = \sqrt{\frac{2}{\pi}} \tag{20}$$

The actual shift of the Daily DVS dataset is 0.75, which aligns closely with the estimated value. However, there remains a slight discrepancy as normalization does not alter the underlying distribution of the original data. If the original data deviates from a normal distribution, the standardized data will not fully conform to normality either. The original data points hold geometric significance, acquired through FPS and KNN methods, yet they do not necessarily adhere to a Gaussian distribution. Given input $[\Delta x, \Delta y, \Delta z, x_c, y_c, z_c]$, we want the network to locate the points within the group by $[x_c + \Delta x, y_c + \Delta y, z_c + \Delta z]$. But we rescale the data, so the expression becomes $[x_c + \Delta x \cdot SD, y_c + \Delta y \cdot SD, z_c + \Delta z \cdot SD]$. When the first three $[\Delta x, \Delta y, \Delta z]$ increase, we want the whole to remain the same, so we should decrease the last three $[x_c, y_c, z_c]$. Consequently, we choose to utilize $[x_{min}, y_{min}, z_{min}]$ rather than precisely correcting the Centroid (centroid $- \sqrt{\frac{2}{\pi}} \times SD$). On the Daily DVS dataset, these two methods achieve testset accuracies of 97.92% and 97.50%, respectively.

## A.2 MRE IN RATE CODING

The error in the transformation process due to the SNN coding using as little as 16 timesteps to describe the 128, 240, and 346 resolution data is not negligible. As a matter of fact, using such a small timestep, within each group of points, the points often become indistinguishable. To better differentiate the points within a group, we apply a rescaling step by Eq. 4 to change the origin and scale of the coordinate system from $d_i$ to $\Delta d_i$. Before and after the rescaling, the Mean Relative Error (MRE) can be expressed by Eq. 21 and Eq. 22, respectively:

$$\delta_{|d_i|} = \frac{1}{n} \sum_{i=1}^n \frac{|\frac{1}{T}\sum_{j=1}^T \chi(|d_i|) - |d_i||}{|d_i|}, \quad |d_i| \in |\mathcal{G} - \text{Centroid}| \tag{21}$$

$$\delta_{\Delta|d_i|} = \frac{1}{n} \sum_{i=1}^n \frac{|\frac{1}{T}SD\sum_{j=1}^T \chi(\frac{|d_i|}{SD}) - |d_i||}{|d_i|}, \quad \Delta|d_i| = \frac{|d_i|}{SD} \tag{22}$$

where $\delta$ denotes MRE, $T$ represents the timesteps in SNN, $SD$ is the standard deviation of $\mathcal{G}$, $n$ indicates the number of points, and $|d|$ refers to the Manhattan distance of each point in a group to the Centroid, where the dimension of $|d|$ is $[N', M, 3]$. We reshape $d$ into one-dimensional to calculate MRE, and $i \in (1, N' \cdot M \cdot 3)$. The stateless Poisson encoder is represented by the function $\chi$.

For instance, in Daily DVS dataset, $SD \approx 0.052$, $|\bar{d}_i| \approx 0.039$, and $\Delta|\bar{d}_i| = \frac{|\bar{d}_i|}{SD} \approx 0.75$. Where $|\bar{d}_i|$ is the average of $|d_i|$ which from all points. As we expected, $\Delta|\bar{d}_i|$ is very close to the theoretical value ($\sqrt{\frac{2}{\pi}}$) we calculate in Eq. 6. Next, through experiment we get $\delta_{\Delta|d_i|} \approx 0.26$ and $\delta_{|d_i|} \approx 1.07$. Our method results in a 76% reduction in encoding error of the coordinates.

### A.3 COEFFICIENT OF VARIATION

In this paper, a stateless Poisson encoder is used. The output $|\hat{d}_i|$ is issued with the same probability as the value $|d_i|$ to be encoded, where $P(|d_i|) = |d_i|$, $|d_i| \in [0, 1]$, and $|\hat{d}_i| = \frac{1}{T} \sum_{i=0}^{T} \chi(|d_i|)$. After performing the rescaling operation, the data, that are not at the same scale, the coefficient of variation (CV) can eliminate the influence of scale and magnitude and visually reflect the degree of dispersion of the data. The definition of CV is the ratio of the standard deviation to the mean, $n$ represents the number of trials, as follows:

$$cv = \frac{SD}{\mu} \quad SD(|\hat{d}_i|) = \sqrt{\frac{(1 - |d_i|)^2 \times |\hat{d}_i| \times n \times T + |d_i|^2 \times (1 - |\hat{d}_i|) \times n \times T}{n \times T}} \tag{23}$$

$$\mu(|\hat{d}_i|) = \alpha|d_i| = \frac{1}{n} \sum_{j=1}^{n} |\hat{d}_i|_j \tag{24}$$

$$cv = \sqrt{\frac{(1 - |d_i|)^2 \times |\hat{d}_i| + |d_i|^2 \times (1 - |\hat{d}_i|)}{(\alpha|d_i|)^2}} = \sqrt{\frac{\alpha|d_i| + |d_i|^2 - 2\alpha|d_i|^2}{(\alpha|d_i|)^2}} \tag{25}$$

Where $\alpha$ is the scale factor, and $\mu(|\hat{d}_i|)$ is an unbiased estimator of the expected value. If the number of trials $n$ is large, then $\alpha$ follows a normal distribution $N(1, \frac{1}{n \cdot \sqrt{T}})$. T is the timestep of SpikePoint, which is 16, $n$ is the number of times the same value is encoded, the input dimension of the network is $[N', M, 6] = [1024, 24, 6]$, the resolution of the value is related to the DVS, for example, Daily DVS is 128, so the average number of trials to each resolution is 1152. We simulated the rate coding with the above data to derive $\alpha \sim N(1.00121, 0.0116)$, which is a narrow normal distribution. Approximately, if we bring $\alpha = 1$ into Eq. 25 to get $cv = \sqrt{1/|d_i| - 1}$. From the formula, it can be concluded that the larger $|d_i|$ is, the smaller the coefficient of variation is, and the network is more easily able to learn the knowledge of the dataset. This improvement is reflected in the accuracy of the trainset of Daily DVS, which increases from 95% before conversion to 99.5% by replacing $|d_i|$ to $\Delta|d_i|$.

### A.4 RESIDUAL IN BACKPROGATION

The residual module in ANN is expressed by the following equation, $A$ is the state vector of the neurons, and $B$ is the bias vector of the neurons:

$$A^l = W^l Y^{l-1} + B^l, \quad Y^l = \sigma(A^l + Y^{l-1}) \tag{26}$$

$$\triangle W_{ij}^l = \frac{\partial L}{\partial W_{ij}^l} = \sigma'(A_i^l) Y_j^{l-1} (\sum_k \delta_k^{l+1} W_{ik}^{T,l} + \sigma'(A_i^{l+m-1} + Y_j^l) \sum_k \delta_k^{l+m} W_{ik}^{T,l+m-1}) \tag{27}$$

When the output of the above formula $A^l$ is 0, the value of $Y^l$ is the value of the input $Y^{l-1}$, $\sigma$ is a nonlinear function. This operation is known as identity mapping. Eq. 27 represents the rule of backpropagation of gradients with residuals, $L$ represents the loss function, $l$ represents the parameters of the $l^{th}$ layer, $\sigma'$ is the derivative of the activation function, $Y^{l-1}$ represents the input of the $l^{th}$ layer of the network, $\delta$ represents the error of output neuron $k$, $m$ represents the output of the current layer after the residuals to the $m^{th}$ layer. If we apply this module directly in SNN, this will cause the problem of gradient explosion or vanishing (Fang et al., 2021a). We can do identity mapping by changing the residual module to the following form in SNN. And the coefficient $\sigma'(I_i^{l+m-1} + S_j^l)$ in Eq. 29 of error propagation of the corresponding residual term is canceled.

$$S^l = \text{LIF}(I + S^{l-1}) \longrightarrow \text{LIF}(I) + S^{l-1} \tag{28}$$

$$\triangle W_{ij}^l = \frac{\partial L}{\partial W_{ij}^l} = \sigma'(I_i^l) S_j^{l-1} (\sum_k \delta_k^{l+1} W_{ik}^{T,l} + \sigma'(I_i^{l+m-1} + S_j^l) \sum_k \delta_k^{l+m} W_{ik}^{T,l+m-1}) \tag{29}$$

$$\sigma(x) = \frac{1}{\pi}\arctan(\pi x) + \frac{1}{2}, \quad \sigma^{'}(x) = \frac{1}{(1 + (\pi x)^2)} \qquad (30)$$

Due to the non-differentiable nature of the $\theta$ function, the surrogate gradient method is often used when training SNN. If $S^{l-1}$ is equal to 1, the expectation of $I_i^{l+m-1} + S_j^l$ is around 1 because $I_i^{l+m-1}$ is the output of batch normalization layer, so the expectation of the surrogate gradient $\sigma^{'}(I_i^{l+m-1} + S_j^l)$ which is calculated by Eq.16 is always less than one. This will cause the error propagated through the residual to become smaller and smaller until it disappears. The modified residual connection can effectively alleviate this phenomenon.

## A.5 EXPERIMENT DETAILS

We evaluate the performance of SpikePoint on our server platform, during both training and testing. Our experiment focuses on measuring the accuracy of the model and examining the size of model parameters across different datasets. The specific specifications of our platform include an AMD Ryzen 9 7950x CPU, an NVIDIA Geforce RTX 4090 GPU, and 32GB of memory. For our implementation, we utilize the PyTorch (Paszke et al., 2019) and spikingjelly (Fang et al., 2020) frameworks.

Table 10: More details in five action recognition datasets.

| DataSet | Resolution | Samples | Record | Denoise | SpikePoint |
|---|---|---|---|---|---|
| DVS128 Gesture | 128x128 | 1342 | direct | × | large |
| Daliy DVS | 128x128 | 1440 | direct | × | small |
| DVS Action | 346x260 | 450 | direct | ✓ | small |
| HMDB51-DVS | 320x240 | 6776 | convert | × | large |
| UCF101-DVS | 320x240 | 13320 | convert | × | large |

### A.5.1 DATASET

The details of the datasets are shown in Table 9. The DVS 128 Gesture dataset is collected by IBM (Amir et al., 2017), comprising 10/11 gesture classes. It is captured with a DAVIS 128 camera under different lighting conditions with a resolution of 128×128. The Daily DVS dataset (Liu et al., 2021a) contains 1440 recordings of 15 subjects performing 12 daily actions under different lighting and camera positions. It is also recorded with DAVIS 128 camera. The DVS Action dataset (Miao et al., 2019) features 10 action classes with 15 subjects performing each action three times. It is recorded with a DAVIS 346 camera with a resolution of 346×240. The HMDB51-DVS (Bi et al., 2020) dataset includes 51 human action categories recorded with conventional frame-based camera recordings and subsequently converted into events using DAVIS 240 conversion mode. The resolution is 320×240. Each of those datasets has its own application scenario, and all of them are widely used for research and benchmarking in neuromorphic computing. We visualize some samples from the four datasets to understand event-based action recognition better, shown in Fig. 5.

### A.5.2 NOISE

There are two main types of noise in the event data, illumination noise (hot pixels) and background noise. Illumination noise pertains to the noise introduced by changes in ambient lighting conditions during data capture, and background noise is the generated event from the unwanted movement of the background. These two types of noise are represented in all four datasets. For the illumination noise, the DVS Action dataset exhibits noticeable illumination noise, depicted in Fig. 6(a). That noise is likely introduced during the recording, as the illumination noise could be present in certain scenarios. The illumination noise profoundly impairs the effectiveness of the network during random sampling. Consequently, we use the denoising method (Feng et al., 2020) to reduce the noise in the sample and the probability of sampling points from noise. As for the remaining three datasets, their illumination noise is relatively low, so we proceeded directly. As for background noise, HMDB51-DVS indicates a strong presence of background noise. As illustrated in Fig. 6(b), the label is hugging

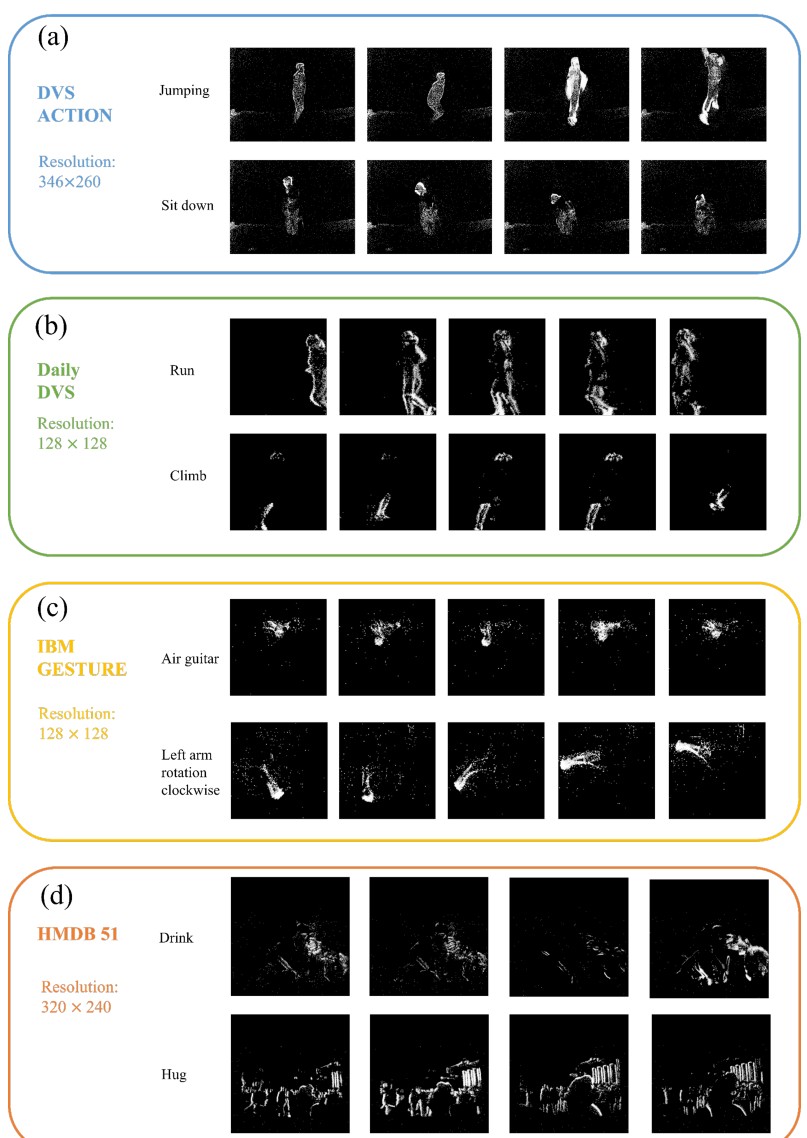

Figure 5: Visualization of four datasets. (a)DVS Action.(b)Daily DVS.(c)IBM Gesture. (d)HMDB51-DVS.

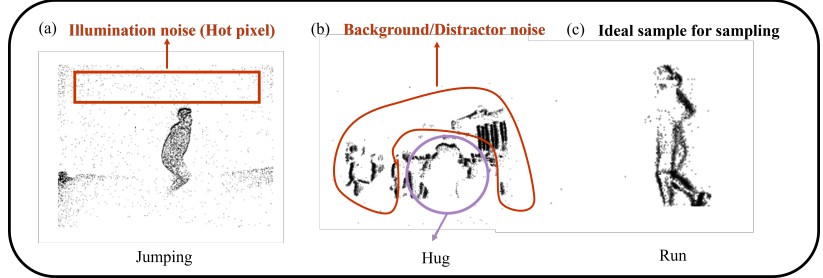

Figure 6: Visualization of different noise. (a) The schematic of illumination noise (b) The schematic of background noise (c) The representation of ideal sample

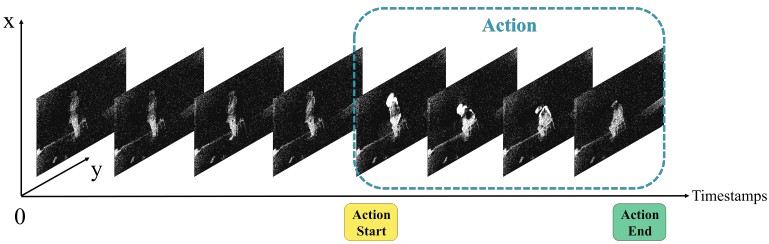

Figure 7: Visualization of a sit sample from DVS Action. We selected the second half of the event stream to build the dataset.

people, and the red area is drawn for the background of the building. Since most of the Point Clouds sent into the network originated from background buildings, the network can not recognize the action. To our best knowledge, this problem remains challenging in the community. The background noise significantly limits the accuracy of the action recognition on the HMDB51-DVS dataset.

### A.5.3 LABEL

In the DVS Action dataset, there is no detailed timestamp setting, only including the action start and end labels. After we visualize the event stream of the files, as depicted in Fig. 7, we find that almost all actions occur after half of the total dataset. So we set the sampling start point from half of the total timestamps to build the dataset. Compared with the manual annotation method (Wang et al., 2020), our method is straightforward but still achieves better results on the DVS Action dataset.

### A.5.4 SLIDING WINDOW

The appropriate length of the sliding window affects the performance of the network. We initially used a sliding window of 0.5 s in Daily DVS, and the accuracy of the model was only 80% with this setting. After comparative experiments, we finally optimized the value with 1.5 s, representing the average length of the 12 actions identified in the Daily DVS dataset. Determining the appropriate length of the sliding window for describing an action is a valuable task to explore in the future, for instance, an automated adjustment window.

### A.6 OVERALL ARCHITECTURE

The pipeline which is shown in Fig. 1 begins with the receipt of the original Point Clouds generated by event cameras. A sliding window clip of the original data in the time axis is performed by employing the method discussed in Section 3.1. Subsequently, a random sampling of the Point Clouds in the sliding window is conducted to create a point set of dimensions $[N, 3]$. The sampling and grouping method highlighted in Section 3.2 is then applied to process the set and derive a group set of dimensions $[N', M, 6]$. The Point Cloud dimension is subsequently

---

**Algorithm 1:** The algorithm of SpikePoint

**Data:** $AR_{raw}$ Event cloud data from event cameras
**Result:** $C$ Categories of action recognition

1 **Clip the event stream;**
2 **while** *not the end of event stream* **do**
3     Obtain the start $t_k$ and end $t_l$;
4     Get the sliding window clip $AR_{\text{clip}}$;
5 **end**
6 **if** $n_{win}$ **then**
7     Randomly sampling from clip ;
8     Convert clip to $PN$ ;
9     Normalized data$\in [0, 1]$ ;
10 **end**
11
12 **Grouping;**
13 Find the Centroid through FPS ;
14 **for** *group* **in** *Centroid* **do**
15     $\mathcal{G} = KNN(\text{Centroid}, N', PN)$;
16     Get $[\Delta|x|, \Delta|y|, \Delta|z|, x_{min}, y_{min}, z_{min}]$;
17 **end**
18
19 **Rate coding** for coordinates,the timestep is **T**;
20 **for** *i in* **T** **do**
21     **Extract features** ;
22     Sequential([**for** block **in** local extractor],[reshape & MP],[**for** block **in** global extractor],[MP]) ;
23     **Classifier**;
24     Sequential($fc_1, bn_1, lif_1, fc_2, bn_2, lif_2$, votinglayer)
25 **end**

---

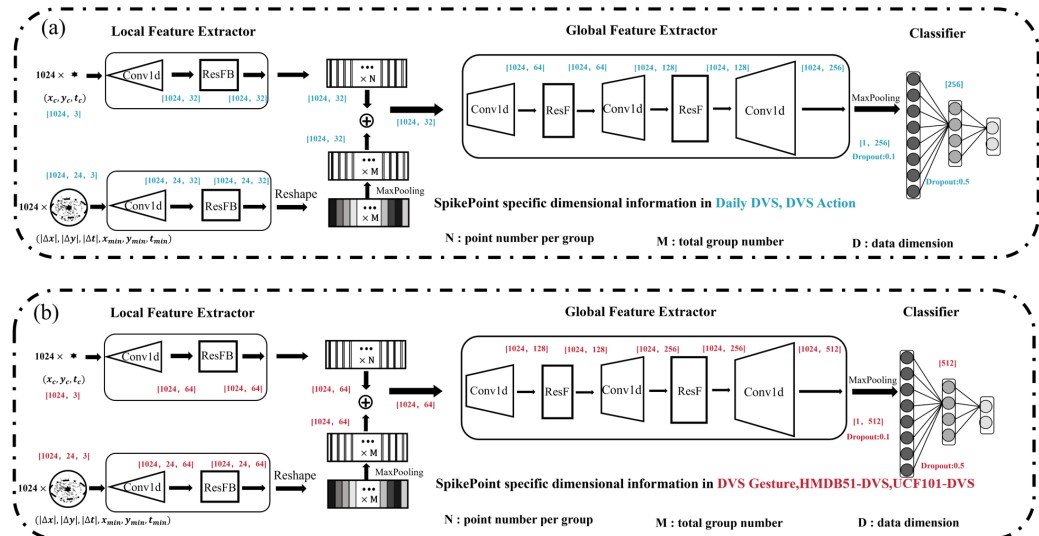

Figure 8: Two models of different sizes corresponding to the four datasets. In addition, each $Conv1D$ is followed by a batch normalization layer, and the bottleneck of $ResFB$ is half of the input dimension.

up-dimensioned by employing a multi-local feature extractor, which results in a tensor of dimensions $[N', M, D]$. The local features are then reshaped and pooled to obtain a tensor of dimensions $[M, D']$, representing the global features. The global features are then extracted utilizing a global feature extractor. The global and local features are abstracted into a single feature vector of dimensions $[1, D'']$ by employing the max pooling operation for classification by the classifier.

## A.7 NETWORK

### A.7.1 STRUCTURE

In this work, we keep the same network architecture for all the datasets. Nevertheless, to accommodate datasets of different sizes, the dimensionality of the network in the feature extraction process is adapted. A relatively small network was used for the Daily DVS and DVS Action, and a relatively large network was used for the other two datasets. The details of the small and large networks are highlighted in blue and red in Fig. 8. We use the list to represent the Point Cloud's dimension change of two different size models, $[32, 64, 128, 256]$ and $[64, 128, 256, 512]$. The first one represents the change in the dimensionality of the input Point Cloud data by the local feature extractor. The remaining three represent the global feature extractor's up-dimensioning of the local feature dimension. In addition, the width of the classifier is $[256, 256, num_{class} * 10]$ and $[512, 512, num_{class} * 10]$ in that order. Due to the smaller dimensions output by the global feature extractor compared to other models, the dropout layer is omitted between the extractor and classifier.

### A.7.2 LIF OR IF

We find that LIF neurons have better performance results on the DVS128 Gesture dataset than IF neurons. They are 98.74% for LIF neurons vs. 97.78% for IF neurons. We speculate that the overfitting of the model is somewhat alleviated due to the leaky of the neurons.

### A.7.3 CLASSIFIER

The classifier aligns with the majority of network solutions by flattening the features and forwarding them to a spike-based MLP with numerous hidden layers for classification. However, instead of the number of neurons in the last layer being equal to the number of categories, a voting mechanism is used to vote with ten times the spike output. Thus, the number of neurons in the classifier's last

layer is ten times the number of categories for classification. We use mean squared error (MSE) as the loss function of SpikePoint. The equation is as follows:

$$\text{Loss} = \frac{1}{T} \sum_{0}^{T-1} \frac{1}{\text{num}} \sum_{0}^{\text{num}-1} (Y - y)^2 \tag{31}$$

The model predicts the result as $Y$ and the actual label as $y$. The encoding timestep of the SNN model is $T$, $num$ is the Action recognition category for the dataset, and the loss function is shown above.

## A.8 HYPERPARAMETERS

We utilize rate encoding to encode coordinates, and the timestep is set to 16. The neuron model we employ is ParamericLIF (Fang et al., 2021b) whose initial $\tau$ is 2.0, and no decay input. In backpropagation, we use the ATan gradient for surrogate training (Neftci et al., 2019).

Our training model uses the following hyperparameters, Batch Size: 6 or 12, Number of Points: 1024, Optimizer: Adam, Initial Learning Rate: $1e - 3$, Scheduler: cosine, Max Epochs: 300.

## A.9 POWER CONSUMPTION DETAILS

We assume that multiply-and-accumulate (MAC) and accumulate (AC) operations are implemented on the 45 nm technology node with $V_{DD} = 0.9\ V$ (Horowitz, 2014), where $E_{\text{MAC}} = 4.6\ pJ$ and $E_{\text{AC}} = 0.9\ pJ$. We calculate the number of synaptic operations (SOP) of the spike before calculating the theoretical dynamic energy consumption by $SOP = firerate \times T \times FLOPs$, where $T$ is the timesteps, and FLOPs is float point operations per sample. OPs in Table 7 refer to SOPs in SNNs and FLOPs in ANNs. The dynamic energy is calculated by $Dynamic = OPs \times E_{\text{MAC or AC}}$ and the static energy is calculated by $Static = Para. \times spp \times L_{\text{sample}}$, where spp is the static power per parameter in SRAM and $L_{\text{sample}}$ is the sample time length, assuming always-on operation. The static power consumption of 1bit SRAM is approximately 12.991 pW in the same technology node (Saun & Kumar, 2019).

