# OpenReview forum: "SpikePoint: An Efficient Point-based Spiking Neural Network for Event Cameras Action Recognition"
_ICLR.cc/2024/Conference — ICLR 2024 spotlight_

### Official Review · Reviewer_tiMk · 2023-10-30

**Soundness:** 3 good
**Presentation:** 2 fair
**Contribution:** 3 good
**Rating:** 6
**Confidence:** 4

**Summary:**

The authors propose a spiking neural network that applies to event-driven camera output and is applied to action detection (agents moving in the visual scene). The authors show that their method achieves performance comparable to state-of-the-art methods, but with significantly lower latency and energy consumption.

**Strengths:**

The method is clearly presented in the paper and is built around the use of point clouds, which are used to represent events. The method uses a relatively classical global architecture that consists in extracting local features, in order to group them intermediately to form a representation that will be efficiently processed by a final classification layer. Overall, the paper is well written and the results are clearly presented.

**Weaknesses:**

A major argument of the paper is to propose a method that deals directly with events that are constituted by the output of an event camera. The authors' argument is to be able to transform events into point clouds and thus improve network performance: "SpikePoint, is an end-to-end point-based SNN architecture". However, the figure shows that after the grouping and sampling stage, the information is transformed by coding the firing rate: "The coordinate is converted into spikes by rate coding, and the results of action recognition are obtained by the local feature extractor, global feature extractor, and classifier in turn". This point needs to be clearly justified, and in particular why isn't the temporal information kept precisely at this point in the processing process. Is that information rather represented in the previous stages?

**Questions:**

In addition, I think the paper could be improved by the following points:

- Numerous methods have been developed in the past to study dynamic scenes, such as particle importance sampling, and in particular the "condensation" method by Isard and Blake. What parallels do you see between your method and these methods?
- In Table 7, you show that performance is optimal for a given number of time steps... What can you deduce from this result in relation to the complexity of the data representation?

Minor:
- "C represents the set of moments" - you mean instants?
- The point "A detailed derivation can be found in Appendix A.4, which describes how this connection solves the problem of backpropagation." is vaguely introduced, please describe minimally the method in the main text.
- The syntax of the paper did not allow me to fully follow all arguments. I have not taken this into account in my evaluation, but the authors should use a service, even an automatic one, that allows clarification of certain points. Fix for instance "bionic neurons" > "biological neurons" or vague statements like "to harmoniously extract local...", . Also check the sentence "We do identity mapping by changing the residual module to the following equation in SNN refer (Hu et al., 2021; Fang et al., 2021a; Feng et al., 2022). And the coefficient σ′ (Il+m−1 + Sl ) in Eq. 29 of error propagation of the corresponding residual term is canceled.
- The LaTeX formatting of the paper could be improved. In particular, quotations in the text should be enclosed in parentheses, e.g. using `citep`. Text appearing in equations ("erf", "clip", "centroid", "lif", ...) should be formatted as text, e.g. using `\text``.

---

> ### Author Response · Authors · 2023-11-15
> **We sincerely  appreciate your recognition of our presentation and your precise summary of SpikePoint.   We will provide detailed answers to your questions and hopefully solve your confusion.**
>
> **Weaknesses  1. The  inconsistency in our arguments that  “deal directly with events that are constituted by the output of an event camera” and the arguments that  “transform events into point clouds and thus improve network performance”.**
>
> These two arguments are not fundamentally contradictory. We apologize for the misuse of the word “convert” and have changed the expression "the event cloud is converted into pseudo-point clouds" to “the event cloud is regarded as pseudo-point clouds” in line 119. In the former argument of “deal directly with events that are constituted by the output of an event camera”, we emphasize that we don't need to stack events through a frame-based method, as this goes against the sparse and asynchronous nature of events. The latter's argument “the event cloud is regarded as pseudo-point clouds”  also maintains that we do not do extra operations with the points. As shown in Section 3.1 in the manuscript, we treat the timestamps of these points as if they were the z-axis of 3D, with no additional transform operations.
>
> **Weakness  2.  SpikePoint is an end-to-end point-based SNN architecture.**
>
> An "end-to-end" architecture, by definition, means that the entire model can be learned and optimized directly from the inputs to the outputs without having to manually design the intermediate steps or features [1].   Processes such as Grouping, Sampling, and Coding require no manual intervention and can seamlessly integrate with subsequent components, including the Local Feature Extractor, Global Feature Extractor, and Classifier. Consequently, SpikePoint necessitates solely the input event cloud and corresponding labels to achieve the complete training of the model.
>
> [1] 3d-siamrpn: An end-to-end learning method for real-time 3d single object tracking using raw point cloud[J]. IEEE Sensors Journal, 2020, 21(4): 4995-5011.
>
> **Weakness 3. Why isn't the temporal information kept precisely at this point in the processing process?  Is that information rather represented in the previous stages?**
>
> Our proposed point cloud processing network fundamentally treats temporal information as a dimension equivalent to 'x' and 'y', as depicted in Eq. 2 in the manuscript. This treatment regards explicit temporal information as an implicit form during the feature extraction process. While lacking the direct temporal context of an RNN, the extracted features inherently encapsulate temporal information through this unique dimensional representation.   Our evaluation results on the five datasets also demonstrate the feasibility and superiority of this approach.  Temporal information ('t') is represented as an implicit feature in the entire model along with spatial information ('x', 'y').
>
> **Q1. What parallels do you see between your method and these methods?**
>
> We express our sincere appreciation for this question and bring the excellent work to our attention! Sampling is a critical process, and it serves as the foundational mechanism for data acquisition and subsequent analysis.
>
> The random sampling employed in our paper for event cloud sampling, as well as the farthest point sampling method utilized in SpikePoint, are characterized by their simplicity and straightforwardness.  We adopt this sampling method the same as PointNet ++ and cite on line 94.
>
> In the Condensation method, factored sampling entails randomly selecting particles from a prior distribution, updating their weights based on observed data, and using the weighted particles as a representation of the posterior density for state estimation. The condensation-based approach based on the probabilistic model,  possesses the capability to effectively track the recognize [2-4].
>
> We believe the condensation method could serve as a potential replacement for our aforementioned sampling methods, thereby augmenting the performance of SpikePoint. We anticipate that this exemplifies how traditional vision and learning-based vision methods can synergistically complement each other. We thank the reviewer again for pointing out this method and we will explore this approach in our future work.
>
> [2] Isard M, Blake A. Condensation conditional density propagation for visual tracking[J]. International journal of computer vision, 1998, 29(1): 5-28.
>
> [3] A probabilistic framework for matching temporal trajectories: Condensation-based recognition of gestures and expressions[C]ECCV'98.
>
> [4] Dynamic gesture recognition by using CNNs and star RGB: A temporal information condensation[J]. Neurocomputing.

---

> > ### Author Response · Authors · 2023-11-15
> >
> > **Q2. What can you deduce from this result in relation to the complexity of the data representation?**
> >
> > Longer timestep yields a more fine-grained data representation, while shorter timestep results in a more coarse representation, albeit with computationally efficient calculations. In the training of SNNs using Backpropagation Through Time (BPTT), a long timestep may lead to inadequate gradient backpropagation, causing issues such as gradient explosion and disappearance. Consequently, despite the finer data granularity, the outcomes may deteriorate. The experimental results prove that 16 is the optimal value for SpikePoint.
> >
> > **Minor 1.**
> > Algorithmically, the set of C represents the instant of spike emission, which leads to the backpropagation dilemma where spikes are not derivable. In this paper, we address this challenge by employing surrogate gradients for training, as detailed in Appendix 4.
> >
> > **Minor 2.**
> > We apologize for keeping this paragraph only in the appendix, due to the length constraints of the main paper. We have added the description on line 208 of the revised paper.
> >
> > **Minor 3&4.**
> > Thank you for proofreading our manuscript. In response to the valuable feedback you provided, all of our authors have thoroughly refined the syntax and improved the LaTeX formatting in the entire paper.
> >
> > Finally, we sincerely hope that our responses have addressed your concerns. Thank you once again for your review and recognition.

---

### Official Review · Reviewer_qCNh · 2023-10-31

**Soundness:** 3 good
**Presentation:** 3 good
**Contribution:** 3 good
**Rating:** 6
**Confidence:** 4

**Summary:**

In this study, the authors present a spiking neural network tailored for event-based action recognition, utilizing event cloud data. The designed network adeptly captures both global and local features. Notably, the introduced method sets new benchmarks by achieving state-of-the-art results on four distinct event-based action recognition datasets.

**Strengths:**

The proposed method is novel and interesting.

The proposed method achieves sota performances on four event-based action recognition datasets.

**Weaknesses:**

1. It is suggested to explain why employing the ResFB in the local extractor and the ResF in the global extractor.

2. Regarding the experiments conducted on DVS Gesture, please specify whether the setting encompasses 10 classes or 11.

3. For clarity in Table 1, it would be more efficient to consolidate all pertinent information within a single row.

4. Could you clarify the term "Single-stream"? Based on Figure 1, the entire network appears to consist of two distinct streams.

5. In the related work section, consider incorporating more contemporary research related to both 'event-based action recognition' and 'point cloud network in ann'.

Minor issues:

There's an inconsistency in the experimental outcomes for SEW-Resnet as presented in Table 2 and Table 6.

**Questions:**

Please refer to 'Weaknesses'.

---

> ### Author Response · Authors · 2023-11-15
> **We sincerely appreciate your comments and your recognition of this paper’s novelty and significance. We would like to address your questions with the utmost diligence and clarity.**
>
> **Weakness 1. It is suggested to explain why employing the ResFB in the local extractor and the ResF in the global extractor.**
>
> Exploring different blocks for local and global extractors is one essential design within our whole architecture. As shown in Fig. 1, the input dimension of local and global extractor differs drastically: the local feature extractor’s input and output dimensions are [M, N’, D]=[1024，24，6], and [M, N’, D1] =[1024, 24, 32], respectively. The global feature extractor’s input and output dimensions are  [M, D2]=[1024，32] and [M， D4]=[1024, 256], respectively. We designed ResFB and ResF for the local and global extractors tailored to accommodate this difference in input and output dimensions of the two feature extractors.
>
> The global feature extractor’s output is directly fed into the classifier, and we need to expand the output dimension and ensure the advanced nature of features to guarantee the effectiveness of the classifier. To expand the output dimension from 32 to 256, we applied 3 Conv1D layers and 2  ResF blocks.
>
> As for the ResFB,  there are a large number of the local feature extractor’s inputs, and these inputs need to go through the shared weight local feature extractor N' x M times for extracting the local features. In order to deal with the problem, we introduce the bottleneck structure with a compression Conv1D layer and an expansion Conv1D layer to efficiently extract features while minimizing computational resources.
>
> **Weakness 2. Regarding the experiments conducted on DVS Gesture, please specify whether the setting encompasses 10 classes or 11.**
>
> We should have been more clear with our experiments. We use 11 classes for model evaluation, which is consistent with other comparative work such as PLIF, SEW-Resnet, and Spikingformer.
>
> **Weakness 3. For clarity in Table 1, it would be more efficient to consolidate all pertinent information within a single row.**
>
> We apologize for adopting this formatting due to page limitations, we have changed it in the revised version.
>
> **Weakness 4. Could you clarify the term "Single-stream"? Based on Figure 1, the entire network appears to consist of two distinct streams.**
>
> The mainstream ANN-based Point Cloud methods often utilize a hierarchy structure with multiple stages. Whereas SpikePoint has only one stage. A stage contains both global and local feature extraction, first seen in  PointNet++, the pioneer of point cloud networks, in the paper named Set Abstraction (SA). Additionally, the hierarchy structure with many stages is not readily applicable to SNNs and often performs inferior compared to the singular stage. This is because spike-based features tend to become sparse and indistinguishable as the depth of the stage increases and the training method based on backpropagation has a serious gradient problem.
>
> **Weakness 5. In the related work section, consider incorporating more contemporary research related to both 'event-based action recognition' and 'point cloud network in ann'.**
>
> We sincerely appreciate your valuable suggestion. In response, we have added the last two years of works in Sections 2.1 and 2.2 of the revised version.
>
> [1] Starting From Non-Parametric Networks for 3D Point Cloud Analysis[C]//Proceedings of the IEEE/CVF Conference on Computer Vision and Pattern Recognition. 2023: 5344-5353.
>
> [2] Pointnext: Revisiting pointnet++ with improved training and scaling strategies[J]. Advances in Neural Information Processing Systems, 2022, 35: 23192-23204.
>
> [3] MPCT: Multiscale Point Cloud Transformer with a Residual Network[J]. IEEE Transactions on Multimedia, 2023.
>
> [4] TTPOINT: A Tensorized Point Cloud Network for Lightweight Action Recognition with Event Cameras[C]//Proceedings of the 31st ACM International Conference on Multimedia. 2023: 8026-8034.
>
> [5] Sparser spiking activity can be better: Feature Refine-and-Mask spiking neural network for event-based visual recognition[J]. Neural Networks, 2023, 166: 410-423.
>
> [6] EventMix: An efficient data augmentation strategy for event-based learning[J]. Information Sciences, 2023, 644: 119170.
>
> **Minor issues: There's an inconsistency in the experimental outcomes for SEW-Resnet as presented in Table 2 and Table 6.**
>
> We apologize for the misunderstanding.  Indeed the SEW-Resnet in Table 2 and Table 6 are from different works. The former has higher accuracy,  but the power consumption is missing. We cite the former in Table 2 to compare the accuracy.  The latter is from [7], they provide the power consumption details. We change the names and make them easily distinguishable.
>
> [7]Going deeper with directly-trained larger spiking neural networks[C]//Proceedings of the AAAI conference on artificial intelligence. 2021, 35(12): 11062-11070.
>
> Thank you once again for your review and for asking highly professional questions. We sincerely hope that our answers have successfully resolved any confusion you may have had.

---

> > ### Comment · Reviewer_qCNh · 2023-11-22
> > **Discussion**
> >
> > The authors' response effectively tackled my concerns. I will maintain my rating as 'weak accept.'

---

### Official Review · Reviewer_NnWo · 2023-10-31

**Soundness:** 2 fair
**Presentation:** 3 good
**Contribution:** 2 fair
**Rating:** 3
**Confidence:** 5

**Summary:**

This paper proposes an SNN framework for event stream processing, termed SpikePoint. It first processes the event stream into point groups and encodes using the rate coding method. Then, the local and global feature extractors are proposed to learn the deep features based on spiking activation neurons.

the writing of this work needs further polishment; a lot of typos can be found all through the paper;
the idea of pure snn for event point stream processing is not new; as the key components are all off-the-shelf modules;
the experiments on large-scale event-based recognition datasets are missing; which is hard to judge whether the proposed method works.

**Strengths:**

This paper proposes an SNN framework for event stream processing, termed SpikePoint. It first processes the event stream into point groups and encodes using the rate coding method. Then, the local and global feature extractors are proposed to learn the deep features based on spiking activation neurons.

**Weaknesses:**

the writing of this work needs further polishment; a lot of typos can be found all through the paper;
the idea of pure snn for event point stream processing is not new; as the key components are all off-the-shelf modules;
the experiments on large-scale event-based recognition datasets are missing; which is hard to judge whether the proposed method works.

**Questions:**

1. further polish this paper;
2. re-organize the contributions of this work, as the current version does not shown significant difference with existing works;
3. more experiments on large-scale event datasets are needed.

---

> ### Author Response · Authors · 2023-11-15
> **We appreciate your reviews and respect the reasons provided for the rejection. We have provided a comprehensive response addressing the weaknesses raised and have revised the manuscript accordingly. We remain hopeful that it could potentially change your decision.**
>
> **Q1. Further polish this paper.**
>
> We sincerely appreciate your feedback on the paper writing. All the authors at SpikePoint have diligently checked and polished it, and we have uploaded the revised version to the rebuttal version.
>
> **Q2. The current version does not show a significant difference from existing works.**
>
> You mentioned that “The idea of pure SNN for event point stream processing is not new.” We agree that there have been many event-processing SNN networks in literature. However, to our best knowledge, all these SNNs are frame-based, resulting in the loss of fine-grained temporal information. We pioneered the application of SNNs to process event streams specifically for Point-based.  The effectiveness of our approach is corroborated by the fact that our results surpass all existing SNN networks. As summarized in the manuscript and also mentioned by other reviewers the novelty includes:
>
> 1. We propose the first SNN-based framework for handling event-based data using the point cloud representation.
>
> 2. We innovate a novel encoding method that effectively captures and encodes negative values pertaining to the relative positioning within the point cloud.
>
> 3. We introduce a streamlined, singular-stage processing framework designed for the efficient adaptation of SNNs.
> 4. We propose powerful local and global feature extractors that can extract features within and between event groups.
> 5. We performed comprehensive evaluations on different scales of event-based action recognition datasets, and SpikePoint achieved SOTA in SNN on all five datasets.
>
> We should have given a clearer explanation of the SNN architecture and its novelty. We have reorganized the manuscript in the last paragraph of the introduction.
>
> **Q3. More experiments on large-scale event datasets are needed.**
>
> We appreciate your invaluable suggestions. Subsequently, we have conducted additional experiments using the large-scale event-based UCF101-DVS dataset, comprising 101 classifications, with a dataset size of nearly 150k.  The preliminary results are summarized in Table 1 below. We have expanded the manuscript to include the results in Section 4.4.
>
>
> Table 1: SpikePoint's performance on UCF101-DVS
> | Name | Method | Param | Acc |
> | -------- | -------- | -------- | -------- |
> | RG-CNN+Incep.3D  [1]   | ANN     | 6.95M     | 63.2%     |
> | I3D     [2]| ANN     | 12.4M     | 63.5%     |
> | ResNext-50  [3]   | ANN    | 26.05M     | 60.2%     |
> | ECSNet-SES  [4]  | ANN     |  -     | 70.2%     |
> | Res-SNN-18    [5] | SNN    |  -     | 57.8%     |
> | RM-RES-SNN-18 [6]    | SNN     | -     | 58.5%     |
> | **SpikePoint**     | **SNN**     | **1.05M**     | **68.46%**     |
>
>
> As illustrated in Table 1, SpikePoint outperforms the state-of-the-art Spiking Neural Networks with a precision of 68.46%. Notably, it represents the latest advancement in the field, building upon RM-RES-SNN [6].  SpikePoint also outperforms the vast majority of ANN's work and is very close to ANN's SOTA. This large-scale experiment further demonstrates the generalization and effectiveness of SpikePoint. Moreover, great performance is achieved with a parameter of only 1.05M,  9.4% of SOTA SNN highlighting the compactness.
>
> [1]Graph-based spatio-temporal feature learning for neuromorphic vision sensing. TIP 2020.
>
> [2]Action recognition? a new model and the kinetics dataset. CVPR 2017.
>
> [3]Can spatiotemporal 3d cnns retrace the history of 2d cnns and imagenet? CVPR 2018.
>
> [4]Ecsnet: Spatio-temporal feature learning for event camera.TCSVT 2022.
>
> [5]Deep residual learning in spiking neural networks. Neurips 2021.
>
> [6]Sparser spiking activity can be better: Feature refine-and-mask spiking neural network for event-based visual recognition. Neural Networks 2023.
>
> Finally, we would like to express our gratitude for your time and effort once again. We sincerely hope that our answers have successfully resolved any confusion you may have had.

---

> > ### Comment · Reviewer_NnWo · 2023-11-23
> >
> > Thanks for your response to my questions.
> > After reviewing the comments of other reviewers and the responses, I may change my rating to a higher score. However, a question still needs to be considered: if the application on event point other than the event frame novel enough for ICLR?

---

> > > ### Author Response · Authors · 2023-11-23
> > >
> > > We are delighted to have addressed your concerns! Our work pioneers in combining point cloud and SNN to process event data in an effective and efficient manner. To our best knowledge, this is the first SNN-based framework to handle event data using the point cloud representation. The sparsity and asynchrony characteristics of events match the trait of point cloud well. Further, SpikePoint’s architecture makes full use of the sparsity and asynchrony of the event data that is represented faithfully in the point cloud. As a result, experiments verify the advantage of our method in terms of energy efficiency in Table 7 and accuracy on five datasets compared to other works in Table 2-6. The accuracy of our method surpasses all frame-based methods on three action recognition datasets, proving the effectiveness of our method.
> > >
> > > Moreover, our innovation is not limited to adapting the point cloud representation scheme with a spiking neural network. Our innovation also includes introducing a pioneering encoding approach to encoding negative values of the relative position data within the point cloud and designing a streamlined, singular-stage structure for the efficient adaptation of SNN.  We also propose powerful local and global feature extractors to extract features both within and between event groups.
> > >
> > > Thank you again for your valuable feedback to improve the quality of this work and for your reconsideration of the rating.

---

### Official Review · Reviewer_XXP9 · 2023-11-06

**Soundness:** 4 excellent
**Presentation:** 4 excellent
**Contribution:** 4 excellent
**Rating:** 8
**Confidence:** 3

**Summary:**

This paper proposed a novel and efficient network approach for event based action recognition. The network leveraged spike neural network as backbone. The preprocessing of the events includes grouping, sampling and rate coding to feed in spike format. The grouping takes special consideration to avoid asymmetric information pass-through. The proposed approach also has shown improving the mean relative error and coefficient of variation.

The SNN learns from both the point cloud centroids and the processed representations. The feature learning part contains both local and global feature extractors as well as residual connection to avoid weight explosion/vanishing.

The approach has been tested on various datasets including small and large ones. The paper has also compared with SOTA methods for similar tasks.

The proposed approach has significantly low power consumption, especially compared to other non SNN based networks. The results are strong and the advantages are salient.

**Strengths:**

The paper has proposed several novel processing steps accompanied by theoretical derivations. The paper first looked at how to convert the events into SNN acceptable format. One of the issues is that directly normalizing delta positions will result in asymmetric information passthrough. The paper calibrated this offset by using the delta of the absolute values. In the SNN part, the paper incorporated residual learning modules to prevent weight explosion/vanishing.

The performance of the proposal has been demonstrated on several datasets and has strong improvement over existing methods.

**Weaknesses:**

I don't find notable weaknesses. I only find the proposed methods could also be extended to other relevant tasks, which this paper has deferred to future work. Otherwise, I think the paper results are pretty solid.

**Questions:**

None.

---

> ### Author Response · Authors · 2023-11-15
> **We sincerely appreciate your recognition of spikepoint's methods, experiments, and performance!**
>
> As a powerful tool to process event data, we believe SpikePoint could extend to more application scenarios. We are endeavoring to broaden the application scope of SpikePoint. We are still working on several applications, such as camera pose relocalization, eye-tracking, and object detection.  Also, SpikePoint also has the potential in low-level event-based tasks such as deblur. Our long-term goal is to run demonstration with our point cloud-based methods on real applications with ultra-low-power systems.
>
> Finally, we extend our heartfelt gratitude once again for acknowledging our research. If you have any new questions or suggestions, we will respond and discuss them with you in a prompt manner.

---

### Author Response · Authors · 2023-11-15

We extend our sincere appreciation to the four reviewers for their valuable and high-quality reviews. Your constructive insights have been instrumental in refining our experiments and enhancing the overall structure of SpikePoint, resulting in a notable improvement in its quality.

We are delighted to highlight the unanimous recognition from all reviewers regarding SpikePoint’s pioneering approach, which seamlessly integrates point cloud and SNN. The accolades of “solid,”“novel and interesting,” “clearly presented,” and “well written” bestowed upon our work truly reflect its groundbreaking nature.

Furthermore, we conducted experiments on a large-scale action recognition dataset and thoroughly checked and refined the entire paper with all authors. We have highlighted the important changes in the revised manuscript. We would like to express our gratitude to the reviewers for their valuable feedback on our paper again.

---

### Comment · Area_Chair_6rfU · 2023-11-15
**Please engage in reviewer-author discussion**

Dear reviewers,

The paper got diverging scores. The authors have provided their response to the comments. Could you look through the other reviews and engage into the discussion with authors? See if their response and other reviewers' comments change your assessment of the submission?

Thanks!
AC

---

### Meta-Review · Area_Chair_6rfU · 2023-12-07

**Metareview:**

This work presents an end-to-end point-based Spiking Neural Network (SNN) architecture for event-based action recognition. It can effectively process sparse event point cloud data with SNN and extract both global and local features with a streamlined singular structure. The proposed SpikePoint achieves state-of-the-art performance on four event-based action recognition datasets, with fewer parameters and more efficient computation.

Strengths
The reviewers generally found the method to be novel and interesting in its approach to handle event data using point cloud representation and further processing with SNN.
The proposed method shows competitive performance in terms of accuracy, model size and efficiency.

Weaknesses
The explanation and presentation could be further improved.

**Justification For Why Not Higher Score:**

Regarding event data as point cloud data for processing borrows much from point cloud processing, which is not that novel.  The explanation and presentation could be further improved.

**Justification For Why Not Lower Score:**

The reviewers generally found the method to be novel and interesting in its approach to handle event data using point cloud representation and further processing with SNN, which is validated through extensive experiments.

---

### Decision · Program_Chairs · 2024-01-16

Accept (spotlight)